# Prescription of oral antidiabetics in Mexico. A cross-sectional study

Inmaculada Fierro[1], Osiel Gallardo-Mora[2], Adela Alba-Leonel[3], Alfonso Carvajal[4], Juan Arcadio Molina-Guarneros[5]*

1 Departamento de Ciencias de la Salud, Universidad Europea Miguel de Cervantes, Valladolid, Spain, 2 Programa de Maestría y Doctorado en Ciencias Médicas, Odontológicas y de la Salud, Universidad Nacional Autónoma de México (UNAM), Mexico, Mexico, 3 Escuela Nacional de Enfermería y Obstetricia, Universidad Nacional Autónoma de México (UNAM), Mexico, Mexico, 4 Universidad de Valladolid, Valladolid, Spain, 5 Departamento de Farmacología, Facultad de Medicina, Universidad Nacional Autónoma de México (UNAM), Mexico, Mexico

* jamg@unam.mx

**Data Availability Statement:** The data are available at https://data.mendeley.com/datasets/mjjcvhdxk6/1.

**Funding:** The author(s) received no specific funding for this work.

## Abstract

In 2016 diabetes was declared an epidemic and a health emergency in Mexico. As the rationale of the treatment is to achieve target glycemia levels, the appropriateness of the medications used is important. The aim of this study is to learn the pattern of antidiabetic drug prescription and factors associated with inappropriate prescription in Mexico. A retrospective cross-sectional drug utilization study has been conducted. A randomly selected sample was carefully examined. Out of 3600 clinical records of patients diagnosed with type 2 diabetes mellitus (T2DM), 196 records were revised. As far as control is concerned, 36.7% had their glycemia values in the recommended range. A combination of different antidiabetics was the most common pattern observed (60.7%); the most frequent was that of the association of metformin with whatever oral antidiabetics. Prescriptions were considered as inappropriate in 149 cases (76.0%); younger age and lack of nutritional assessment was significantly related to inappropriate prescription. A trend to use more drugs for treating T2DM has been consistently observed. Despite using so many drugs, most of the patients are not controlled. Avoiding inappropriate prescription by following current guidelines may contribute to a better control and, in turn, decrease morbidity and mortality for this cause.

## Introduction

Type 2 diabetes mellitus (T2DM) is a metabolic disease largely characterized by impaired insulin secretion and action. It accounts for a public health burden associated with enormous health care and societal costs, early death, and morbidity; it gives rise to complications such as heart and vascular disease, renal disease, or neuropathies. Currently, it has become a global pandemic due to the increasing longevity and prevalence of contributing factors, such as obesity [1]; in 2016, diabetes was declared an epidemic and a national health emergency in Mexico [2].

Lifestyle changes should be the initial approach to diabetes management, and these include dietary interventions, weight reduction, tobacco cessation, physical exercise, and stress management. However, pharmacological treatment of T2DM is, in time, the most common

**Competing interests:** The authors have declared that no competing interests exist.

intervention; it has been consistently shown to reduce the relative risk of cardiovascular mortality and morbidity, neuropathies, and total mortality [3]. Though prevalence of the disease has increased elsewhere, in Mexico the National Health and Nutrition Survey (ENSANUT) reported a prevalence of diabetes in the population over 20 years old of 9.2% in 2012, being 10.3% in 2018 [4]; the figure reaches 30% in the population over 50 years old [5].

Regarding pharmacological treatment, several studies in different countries have identified its inappropriate use in significant proportions [6, 7]. In a large survey carried out in two neighborhoods in Mexico City between 1998–2004 and 2015–2019, there was an increased use of glucose-lowering medications mainly due to a large increase of metformin [8]; in this sample, the proportion of people with diabetes taking at least two of these medications increased from 13% in 1998–2004 to 42% in 2015–2019. In addition to this increase, some studies have similarly identified different degrees of inappropriate prescriptions [9–11]; in the last study carried out in a family medical unit of social security from Comalcalco, Mexico, during 2013, inappropriate prescription of oral antidiabetics was found in 141 patients out of a sample of 150 patients analyzed [11].

As the rationale of the treatment is to achieve target glycemia levels, the type and number of oral antidiabetics along with their appropriateness is important. Therefore, our purpose is to further learn the pattern of oral antidiabetic drug prescription, and to explore factors associated with inappropriate prescription of those drugs in an urban population in Mexico.

## Methods

For the purpose, a retrospective cross-sectional drug utilization study has been conducted. A randomly selected sample of medical records from Ixtapaluca regional hospital (Hospital Regional de Alta Especialidad de Ixtapaluca, HRAEI) was carefully examined. The sample comprised records corresponding to January 2014 to November 2018 of those patients over 18 years old diagnosed with T2DM (E11, CIE-10), who were seen in outpatient clinics and prescribed at least an oral antidiabetic; incomplete records and those corresponding to pregnant women were not included. This hospital covers a broad population of around 250,000 inhabitants from different states (Morelos, Puebla, Mexico City and Hidalgo, among others). All relevant information was retrieved, with a common template, by one of the signing authors (OG-M); he is a qualified medical doctor. Particular attention was paid to demographic data, lifestyle habits, investigations, comorbidities, and medications in the last visit to the doctor. Inappropriate prescriptions were considered those not following official guideline basic recommendations [12], i.e., not having a Body Mass Index (BMI) recorded, or no renal function data registered (glomerular filtration rate, GFR), or having some contraindications for the medications. Controlled diabetes was defined by a threshold HbA1c level of 7%.

This project was filed as self-determined activities not regulated as human participant research with the Hospital Regional de Alta Especialidad de Ixtapaluca (HRAEI) institutional Ethics Committee; this institutional review board which approved the study is an appropriately constituted group that has been formally designated to review and monitor biomedical research involving human subjects and it is also formally registered in the Mexican official list for investigational ethics committees. Given the characteristics of the study based on anonymized retrospective data, the Committee, bearing in mind Mexican legislation on research with human beings and on data protection, did not consider the explicit exemption of informed consent necessary. Nevertheless, the Hospital Ethics Committee authorized and oversaw data anonymization. The study meets the requirements of the current Mexican law regarding health research [13]; it is also in accordance with the international relevant guidelines and regulations.

## Statistics

Most of the results are expressed in frequencies and percentages. Quantitative variables are expressed as mean (standard deviation) or median [interquartile range]. The comparison of the age between sexes is performed by the independent samples t-test. Spearman´s rank correlation coefficient (Rho) is used to analyze correlation between non-normal distributed variables. Odds ratios (OR) are used as association measures between each variable of interest and inappropriate prescription. A logistic regression model was built to know the main factors associated with inappropriate prescription; the goodness of fit was assessed by the Hosmer–Lemeshow test.

The data have been analyzed with the statistical software IBM SPSS v.28

## Results

Out of 3600 clinical records of patients diagnosed with T2DM in the period and setting considered, 196 records were finally revised; Table 1 presents the main characteristics of those patients. Women in the sample were more numerous than men (71.9% vs. 28.1%) and the age was lower in women—the difference was not statistically significant [mean, 54.93 years (13.07) vs. 57.05 years (12.79); p = 0.305]. As far as control is concerned, of those having information (n = 188), 36.7% had their glycemia values in the recommended therapeutic range.

Far from the rest of oral antidiabetics, metformin was the most frequently used (Table 2). A combination of different antidiabetics was present in 60.7% of the cases; the most common pattern encountered was that of the association of metformin with whatever oral antidiabetic. Insulin was used in 41 different cases always as a complementary medication (21%). The median number of whatever antidiabetics per patient was 2.0 [1.0–2.0]; the median number of whatever medications per patient was 5.0 [3.0–6.75].

Prescriptions were considered as inappropriate in 149 cases (76.0%) (Table 3); there were no registered estimates of GFR in 129 cases (65.8%), no BMI in 17 cases (8.7%), and there were contraindications for the prescribed medication in 24 cases (12.2%). Younger age and lack of nutritional assessment were significantly related to inappropriate prescription; age was correlated with time from diagnosis (Spearman rho = 0.429; p<0.001)

## Discussion

The main characteristics of the randomly selected sample we have studied are quite coincidental with other recent T2DM samples in Mexico [14]; women outnumber men by a factor of 2:1. The median BMI corresponds with overweight according to the WHO criteria; in fact, nearly 85% of the patients had overweight or obesity (overweight, 40.8%; obesity, 43.6%). Of particular importance is the elevated number of patients with glycemia levels out of the therapeutic targets, reaching almost two-thirds in this sample; this has been similarly observed in recent surveys in Mexico [4, 14]. Regarding to the pattern of use of antidiabetics, it follows that already described [8]; metformin appears as the major oral antidiabetic and there is a trend to use these drugs in combination: more than half of metformin and almost the rest of the antidiabetics are prescribed in this form, and only a small but significant proportion received insulin. As reported by Aguilar-Ramirez et al. [14], the proportion of people with T2DM taking at least two glucose-lowering medications increased from 13% in 1998–2004 to 42% in 2015–2019; our figures of 60.7% for a period ranging from 2014 to 2018 are somehow higher but confirm the trend of using these drugs in combination. Since the lack of glycemia control is quite remarkable, it is reasonable to wonder the importance of using so many drugs.

Worthy of note is the elevated percentage of inappropriate prescriptions, which accounts for more than 3 out of 4 treated patients. Inappropriate prescription has attracted attention

**Table 1. Main characteristics of the sample studied (n = 196).**

| | | Frequency (%) [a] |
|---|---|---|
| Age (years) | | 56.0 [48.0–64.0] |
| Sex | Men | 55 (28.1) |
| | Women | 141 (71.9) |
| Education level [b] | Lower level | 73 (37.2%) |
| | Higher level | 46 (23.5%) |
| | Missing | 77 (39.3%) |
| Alcohol intake | Yes | 55 (28.1) |
| | No | 122 (62.2) |
| | Missing | 19 (9.7) |
| Smoking | Yes | 48 (24.5) |
| | No | 131 (66.8) |
| | Missing | 17 (8.7) |
| Polypharmacy [c] | Yes | 105 (53.6) |
| | No | 91 (46.4) |
| BMI (kg/m$^2$) [d] | | 28.9 [26.3–33.3] |
| Time from diagnosis (years) | | 6.0 [3.0–14.8] |
| Insulin | Yes | 41 (20.9) |
| | No | 155 (79.1) |
| Control | Yes | 69 (35.2) |
| | No | 119 (60.7) |
| | Missing | 8 (4.1) |
| Nutritional assessment | | 97 (49.5) |
| Physical activity assessment | | 77 (39.3) |
| Medical specialty | Endocrinology | 38 (19.4) |
| | Others [e] | 158 (80.6) |

[a] n (%) or median [Q1-Q3]

[b] Education. Lower: illiterate and primary studies; higher: secondary and superior.

[c] Polypharmacy, defined as 5 or more medications including antidiabetics.

[d] Body Mass Index (BMI) calculated upon 179 patients having this information. Distribution according to standard BMI categories: underweight, 2 (1%); normal weight, 26 (14.5%); overweight, 73 (40.8%); obesity, 78 (43.6%)

[e] Others different from endocrinology

elsewhere in the last decades [15–18]; in the present study, we have identified two factors presumably associated with this practice, i.e., young age and the lack of nutritional assessment. Age could be regarded as correlated with duration of the disease and, in fact, this is what we have observed in our sample; thus, in the long run, there would be more probabilities to carry out the required examinations for a prescription to be appropriate (BMI and GFR). Additionally, there exists the possibility that doctors pay more attention to elderly and comorbid patients, which is consistent and in line what has been observed in the multivariant and bivariant analysis. On its part, nutritional assessment is the cornerstone of these patients' anamnesis, and, in theory, it should precede any treatment. It is conceivable that, when a correct approach from the beginning has been followed, it encompasses the rest of the procedures including examinations and treatment. All in all, since this is a cross-sectional study, we cannot infer any causal association; nevertheless, the associations found contribute to understand prescribing performance. Prescribing does not depend solely on patients' factors; other factors related to the setting and doctors would have an influence as well. We have not identified

**Table 2. Oral antidiabetics.** Pattern of use.

|  | Alone | In combination |
|---|---|---|
|  | Frequency (%) | Frequency (%) |
| Metformin | 71 (38.0) | 116 (62.0) |
| *Insulin* [a] |  | *38 (32.8)* |
| Sulfonylureas [b] | 5 (6.9) | 67 (93.1) |
| *Insulin* [a] |  | *3 (4.5)* |
| Gliptins [c] | 1 (3.0) | 32 (97.0) |
| *Insulin* [a] |  | *11 (34.4)* |
| Other [d] | 0 (0) | 3 (100) |
| *Insulin* [a] |  | *1 (33.3)* |
| Corresponding patients | 77 (39.3) | 119 (60.7) |

[a] Including insulin; % in here refers to combination schemes

[b] Glibenclamide, 70; glimepiride, 2

[c] Sitagliptin, 17; linagliptin, 15; alogliptin, 1

[d] Pioglitazone, 1; dapagliflozin, 2

medical specialty with inappropriate prescribing; no differences were found when comparing endocrinologists to other specialists.

Diabetes in Mexico is not only a highly prevalent condition but a leading cause of death, [19–21]; it accounted for 13.6% of the registered deaths in 2021 [22]; 90% of them attributed to type 2 diabetes. Since the prevalence of diabetes in Mexico is among the highest in the world [14], causing approximately one-third of all premature deaths in adults [23] and much disability and expense [24], increasing attention is being paid to this problem. Regardless the multifactorial approach needed, medication and its appropriate usage is paramount. Studies around the world have explored the correctness of use of glucose-lowering drugs; all these studies point out to an inappropriate prescribing, as much in Mexico [9–11] as in the rest of the world [25–28], of these medications, which coincides with findings of the present study.

The present study has some limitations. Sample size could be regarded as scarce and restricted to a region; in fact, the sample analyzed of 196 individuals would be in between that needed to detect an inappropriate percentage of 76.0% in this population, with an acceptable margin of error of 5% and a confidence level of 90% (n = 187), and the number needed to reach a confidence level of 95% (n = 259). The results put into perspective are, as previously stated, quite like those coming from broader surveys referred to the whole country. On the other hand, clinical records are not intended for research, the information included in these files does not fulfill the stringent requirements of a study protocol; besides, it would be possible that some data or examinations, such as BMI or GFR, had been calculated but not included. Not performed or not included in the clinical records could likewise lead to inappropriate prescribing. Validated criteria to assess inappropriate prescriptions in T2DM have not been developed so far [29]; however, criteria used in this study are quite consistent with those used elsewhere. Since there is no clear information on adherence, prescribing does not mean taking the medication; in addition, several drugs, when associated with lack of control, may be a risk marker more than a risk factor.

In summary, a trend to use more drugs for treating T2DM has been consistently observed. Despite using so many drugs, a number of patients, reaching two-thirds, are not controlled and their glycemia values were out of the therapeutic range. A clinical practice for managing T2DM in line with current guidelines may avoid inappropriate prescription and improve

**Table 3. Oral antidiabetics.** Factors associated with inappropriate prescription.

| | Inappropriate (n = 149) Median [Q1-Q3] or n (%) | Appropriate (n = 47) Median [Q1-Q3] or n (%) | OR [CI 95%] | p | Adjusted OR [CI 95%][a] | p |
|---|---|---|---|---|---|---|
| Age | 54.0 [45.5–64.0] | 58.0 [52.0–66.0] | 0.97 [0.94–0.996] | 0.023 | 0.96 [0.93–0.99] | 0.006 |
| Sex (men) | 38 (25.5) | 17 (36.2) | 0.60 [0.30–1.22] | 0.158 | | |
| Time from diagnosis (years) | 6.0 [3.0–15.0] | 9.0 [4.0–14.3] | 0.99 [0.95–1.03] | 0.554 | | |
| Nutritional assessment | 67 (45.0) | 30 (63.8) | 0.46 [0.24–0.91] | 0.026 | 0.40 [0.20–0.83] | 0.014 |
| Physical activity assessment [b] | 52 (35.1) | 25 (54.3) | 0.46 [0.23–0.89] | 0.021 | | |
| Smoking | 33 (24.3) | 15 (34.9) | 0.60 [0.29–1.25] | 0.173 | | |
| Alcohol intake | 38 (28.4) | 17 (39.5) | 0.61 [0.30–1.24] | 0.170 | | |
| Comorbidity | 128 (85.9) | 46 (97.9) | 0.13 [0.02–1.01] | 0.051 | | |
| Polypharmacy [c] | 75 (50.3) | 30 (63.8) | 0.57 [0.29–1.13] | 0.108 | | |
| Other medications [d] | 128 (85.9) | 46 (97.9) | 0.13 [0.02–1.01] | 0.051 | | |
| Several antidiabetics | 90 (60.4) | 29 (61.7) | 0.95 [0.48–1.86] | 0.874 | | |
| Metformin | 145 (97.3) | 42 (89.4) | 4.32 [1.11–16.80] | 0.035 | | |
| Sulfonylureas | 58 (38.9) | 14 (29.8) | 1.50 [0.74–3.05] | 0.259 | | |
| Gliptins | 21 (14.1) | 12 (25.5) | 0.49 [0.22–1.07] | 0.072 | | |
| Insulin | 26 (17.4) | 15 (31.9) | 0.45 [0.21–0.95] | 0.036 | 0.46 [0.21–1.01] | 0.052 |
| Other antidiabetics | 1 (0.7) | 2 (4.3) | 0.15 [0.01–1.72] | 0.128 | | |
| Doctor specialty / endocrinologist [e] | 29 (19.5) | 9 (19.1) | 1.02 [0.44–2.35] | 0.962 | | |

[a] Hosmer–Lemeshow test, 4.769; p = 0.782

[b] Two missing values for this variable (inappropriate n = 148; appropriate n = 46)

[c] Polypharmacy, 5 or more medications

[d] Having other medications, one or more medications other than antidiabetics.

[e] Endocrinologists vs. other specialists

medical attention; it also may contribute to a better glycemia control in these patients. Control, in turn, is related to a lesser morbidity and mortality burden for this cause. More specifically, paying attention to nutritional anamnesis might be of particular interest according to what was observed in the present study.

## Acknowledgments

Our appreciation to the Management of the Hospital Regional de Alta Especialidad de Ixtapaluca (HRAEI) in Mexico for facilitating the realization of this study. Our special recognition to Dr. Eliseo Pérez Silva, Head of Internal Medicine Unit where this study was performed and to

Dr. Ema Hernández and the rest of the Clinical Analysis Laboratory staff, all from the same institution (HRAEI).

## Author Contributions

**Conceptualization:** Alfonso Carvajal, Juan Arcadio Molina-Guarneros.

**Data curation:** Osiel Gallardo-Mora, Adela Alba-Leonel.

**Methodology:** Inmaculada Fierro, Alfonso Carvajal, Juan Arcadio Molina-Guarneros.

**Software:** Inmaculada Fierro.

**Writing – original draft:** Inmaculada Fierro, Osiel Gallardo-Mora, Adela Alba-Leonel, Alfonso Carvajal, Juan Arcadio Molina-Guarneros.

**Writing – review & editing:** Inmaculada Fierro, Osiel Gallardo-Mora, Adela Alba-Leonel, Alfonso Carvajal, Juan Arcadio Molina-Guarneros.

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
