## [Decision Letter · Decision Letter 0]

10 Apr 2023

PONE-D-23-03248Prescription of oral antidiabetics in Mexico. A cross-sectional studyPLOS ONE

Dear Dr. Fierro,

Thank you for submitting your manuscript to PLOS ONE. After careful consideration, we feel that it has merit but does not fully meet PLOS ONE’s publication criteria as it currently stands. Therefore, we invite you to submit a revised version of the manuscript that addresses the points raised during the review process.

We look forward to receiving your revised manuscript.

Kind regards,

Tariq Jamal Siddiqi

Academic Editor

PLOS ONE

Journal Requirements:

Prescription patterns of antihypertensives in a community health centre in Mexico City: a drug utilization study - https://doi.org/10.1111/fcp.12179

In your revision ensure you cite all your sources (including your own works), and quote or rephrase any duplicated text outside the methods section. Further consideration is dependent on these concerns being addressed.

Reviewers' comments:

Reviewer's Responses to Questions

**Comments to the Author**

1. Is the manuscript technically sound, and do the data support the conclusions?

Reviewer #1: Yes

Reviewer #2: No

2. Has the statistical analysis been performed appropriately and rigorously? 

Reviewer #1: Yes

Reviewer #2: No

3. Have the authors made all data underlying the findings in their manuscript fully available?

Reviewer #1: Yes

Reviewer #2: No

4. Is the manuscript presented in an intelligible fashion and written in standard English?

Reviewer #1: Yes

Reviewer #2: No

5. Review Comments to the Author

Reviewer #1: Fierro et al conducted a retrospective cross-sectional study titled: “Prescription of oral antidiabetics in Mexico. A cross-sectional study” in which they evaluated the pattern of antidiabetic drug prescription and factors associated with inappropriate prescription in Mexico. In my opinion, this study can be improved by inculcating the following changes into the manuscript:

1. It would improve the quality of the manuscript if the authors briefly summarize the existing literature on this topic, identify the gap in knowledge, and mention how their study aims to fill that gap in knowledge in the introduction section of the manuscript.

2. Methods section, page 5, lines 107-110: though the authors have mentioned what information was retrieved from the records, it would improve the manuscript, if the authors briefly mention who retrieved the information, their respective qualifications.

3. Methods section, page 5, lines 118-124: though the authors have mentioned that the study did not require informed consent, and that the study meets the requirements of current Mexican law regarding health research, it would add to the quality of the manuscript if the authors mention a few lines regarding the institutional review board (IRB) approval as well.

4. Though the authors have explained and compared the findings of their study with the previous researches, it would add to the significance and relevance of the study, if the authors mention the clinical implications of their findings, how their findings expand on the existing literature, the future research implications, in the discussion section of the manuscript.

5. The authors should consider, thoroughly proofreading the manuscript to rectify the typos and grammatical errors present at certain places in the manuscript.

Reviewer #2: Thank you for submitting your article for consideration to PloS one. Given the ongoing epidemic of uncontrolled T2DM in Mexico. This study highlights a vital cause of inappropriate prescription use that contributes towards uncontrolled T2DM. Despite the issues raised, the article needs a re-write with appropriate grammatical construction and scientific language.

Line 147-150 talks about the glycemic values in the control population. 36.5% had within recommended range and then 85% had uncontrolled DM. The author needs to mention strategies of how to improve diabetes control in the discussion.

6. PLOS authors have the option to publish the peer review history of their article (what does this mean?). If published, this will include your full peer review and any attached files.

Reviewer #1: No

Reviewer #2: **Yes: **Mahammed Ziauddin Khan suheb

---

## [Author Response · Author response to Decision Letter 0]

28 May 2023

Editor

1. “Please ensure that your manuscript meets PLOS ONE's style requirements […]”. We have checked the manuscript to meet the journal´s requirements.

2. “We noticed you have some minor occurrence of overlapping text with the following previous publication(s), which needs to be addressed: Prescription patterns of antihypertensives in a community health centre in Mexico City: a drug utilization study - https://doi.org/10.1111/fcp.12179”. There is in fact an overlapping text: “For the purpose, a retrospective cross-sectional drug utilization study has been conducted” and “Most of the results are expressed in frequencies and percentages”. They have previously appeared in the Methods section of a previous article by our group; they have a general character and do not affect the originality of the present article. There is no duplicate text outside the methods section.

3. “Please provide additional details regarding participant consent. In the ethics statement in the Methods and online submission information, please ensure that you have specified (1) whether consent was informed and (2) what type you obtained (for instance, written or verbal, and if verbal, how it was documented and witnessed). If your study included minors, state whether you obtained consent from parents or guardians. If the need for consent was waived by the ethics committee, please include this information.

If you are reporting a retrospective study of medical records or archived samples, please ensure that you have discussed whether all data were fully anonymized before you accessed them and/or whether the IRB or ethics committee waived the requirement for informed consent. If patients provided informed written consent to have data from their medical records used in research, please include this information.” According to your suggestion, we have re-written the “ethics statement” in the Methods section as follows, “This project was filed as self-determined activities not regulated as human participant research with the Hospital Regional de Alta Especialidad de Ixtapaluca (HRAEI) institutional Ethics Committee; this institutional review board is an appropriately constituted group that has been formally designated to review and monitor biomedical research involving human subjects, and it is also formally registered in the Mexican official list for investigational ethics committees. All data were fully anonymized before we accessed them.

“In your Data Availability statement, you have not specified where the minimal data set underlying the results described in your manuscript can be found […]

We have made data available and we have introduced new information: The data are available at https://data.mendeley.com/datasets/mjjcvhdxk6/1

Reviewer 1

1. Introduction section. It is recommended [to] “summarize the existing literature on this topic, identify the gap in knowledge and mention how their study aims to fill that gap in knowledge in the introduction section of the manuscript”. There is current general agreement about the shortage of studies addressing the patterns of use of antidiabetics and its correctness in Mexico (Corona-Rojo JA et al. Potential prescription patterns and errors in elderly adult patients attending public primary health care centers in Mexico City. Clin Interv Aging. 2009;4: 343-50. https://doi.org/10.2147/cia.s5198). In the submitted manuscript, we identified and referenced two published studies upon “inappropriate prescription” (Mino-León D et al. Potentially inappropriate prescribing to older adults in ambulatory care: prevalence and associated patient conditions. Eur Geriatr Med. 2019;10: 639–647; Zavala-González MA et al. Utilización de hipoglucemiantes orales en una unidad médica 392 familiar de Comalcalco, Tabasco, México, 2013. Rev. Mex. Cien. Farm. 2014;45: 81-85); they were quoted in the Discussion section of the manuscript. As suggested, a new search has been conducted identifying a new study that we have accordingly incorporated to the manuscript (Cardoza-Contreras D et al. Inappropriate Drug Prescription in the Elderly in Tijuana. J Fam Med, 2019, vol. 6, no 2, p. 1161). To summarize the existing literature, we have quoted the three studies and briefly described them in the Introduction section regardless their quotation in the Discussion section. 

2. Methods section (page 5, lines 107-110). It is stated that, “though the authors have mentioned what information was retrieved from the records, it would improve the manuscript, if the authors briefly mention who retrieved the information, their respective qualifications”. Osiel Gallardo-Mora (OG-M), one of the signing authors of this manuscript, retrieved the information from the records. We have accordingly introduced this information in the revised manuscript as follows: “All relevant information was retrieved, with a common template, by one of the signing authors (OG-M); he is a qualified medical doctor”.

3. Methods section (page 5, lines 118-124). “…though the authors have mentioned that the study did not require informed consent, and that the study meets the requirements of current Mexican law regarding health research, it would add to the quality of the manuscript if the authors mention a few lines regarding the institutional review board (IRB) approval as well”. The Hospital Regional de Alta Especialidad de Ixtapaluca (HRAEI) IRB is an appropriately constituted group that has been formally designated to review and monitor biomedical research involving human subjects; in accordance, it is formally registered (number 127) in the Mexican official list for investigational ethics committees (https://www.gob.mx/cms/uploads/attachment/file/442503/Registros_CEI_25022019.pdf). Since another remark upon this point has been posed by the Editor, further information has been introduced in the corresponding section (see responses to questions posed by the editor).

4. Discussion section. It is suggested that “clinical implications of their findings, how their findings expand on the existing literature, the future research implications, in the discussion section of the manuscript” [should be mentioned]. Regarding clinical implications, it is said in the current manuscript that “A clinical practice for managing T2DM in accordance with current guidelines may avoid inappropriate prescription and improve attention; it also may contribute to a better glycemia control in these patients”. Since “nutritional anamnesis” appears to avoid inappropriate prescription, the following statement has been added: “More specifically, paying attention to nutritional anamnesis might be of particular interest according to what was observed in the present study”.

5. General. “…thoroughly proofreading…”. See point 1 posed by Reviewer 1.

Reviewer 2

1. General. It is said that “the article needs a re-write with appropriate grammatical construction and scientific language”. This is coincidental with that suggested by Reviewer 1. The whole manuscript has been thoroughly proofread to rectify the typos and grammatical errors and, when applied, re-written with scientific language.

2. Results section. “Line 147-150 talks about the glycemic values in the control population. 36.5% had within recommended range and then 85% had uncontrolled DM”. It is in fact an error, we do apologize. The first percentage (36.5%), coincidental with figures in Table 1 and in the Abstract, is the correct one; the last one (85%) has been deleted.

3. Discussion section. “The authors need to mention strategies of how to improve diabetes control”. This requirement is in part like that suggested by Reviewer 1. Diabetes control is a target difficult to achieve. In a recently published cross-sectional analysis of data from adults with diabetes in the United States participating in the National Health and Nutrition Examination Survey to assess national trends in diabetes treatment and risk-factor control from 1999 through 2018, the percentage of adult participants with diabetes, in whom glycemic control (glycated hemoglobin level, <7%) was achieved, declined from 57.4% (2007-2010; n=1481) to 50.5% (2015-2018; n=1718). [Fang M et al. Trends in Diabetes Treatment and Control in U.S. Adults, 1999-2018. N Engl J Med. 2021; 384 (23):2219-2228]. Diabetes control implies different stakeholders: health authorities, health professionals and patients themselves; scientific work somehow may help. We have re-written this point as follows, “A clinical practice for managing T2DM in line with current guidelines may avoid inappropriate prescription and improve attention; it also may contribute to a better glycemia control in these patients. More specifically, paying attention to nutritional anamnesis might be of particular interest according to what was observed in the present study”. 

Additional minor amendments. Other amendments have been performed. Some errors not affecting the overall results have been corrected and appropriately indicated in red in the Results section. Similarly, an update in the number of deaths attributed to diabetes in Mexico has been introduced in the Discussion section; the corresponding source has accordingly been referenced (INEGI. Estadísticas de las defunciones registradas en México, 2021). 

All changes have been introduced, in red, in the 'Revised Manuscript with Track Changes.'

---

## [Decision Letter · Decision Letter 1]

13 Jun 2023

PONE-D-23-03248R1Prescription of oral antidiabetics in Mexico. A cross-sectional studyPLOS ONE

Dear Dr. Fierro,

Thank you for submitting your manuscript to PLOS ONE. After careful consideration, we feel that it has merit but does not fully meet PLOS ONE’s publication criteria as it currently stands. Therefore, we invite you to submit a revised version of the manuscript that addresses the points raised during the review process.

We look forward to receiving your revised manuscript.

Kind regards,

Tariq Jamal Siddiqi

Academic Editor

PLOS ONE

Journal Requirements:

Reviewers' comments:

Reviewer's Responses to Questions

**Comments to the Author**

1. If the authors have adequately addressed your comments raised in a previous round of review and you feel that this manuscript is now acceptable for publication, you may indicate that here to bypass the “Comments to the Author” section, enter your conflict of interest statement in the “Confidential to Editor” section, and submit your "Accept" recommendation.

Reviewer #1: (No Response)

2. Is the manuscript technically sound, and do the data support the conclusions?

Reviewer #1: Yes

3. Has the statistical analysis been performed appropriately and rigorously? 

Reviewer #1: Yes

4. Have the authors made all data underlying the findings in their manuscript fully available?

Reviewer #1: Yes

5. Is the manuscript presented in an intelligible fashion and written in standard English?

Reviewer #1: Yes

6. Review Comments to the Author

Reviewer #1: 1. Page 6, methods section, line 134: while the authors have acknowledged that all data was anonymised, it is recommended that they also mention wether the ethics committee granted a waiver for informed consent.

7. PLOS authors have the option to publish the peer review history of their article (what does this mean?). If published, this will include your full peer review and any attached files.

Reviewer #1: No

---

## [Author Response · Author response to Decision Letter 1]

21 Jun 2023

Dear Editor,

Thank you for giving us the opportunity to resubmit the manuscript. We have revised the document according to the comments and suggestions that have been made. Corrections have been introduced as indicated and a marked-up copy of the manuscript that highlights changes made to the original version has been built-up. Changes appear in red.

We do thank you for your work. No doubt, the comments and suggestions provided have contributed to noticeably improve the manuscript.

Sincerely yours,

---

## [Editor Report · Decision Letter 2]

4 Jul 2023

Prescription of oral antidiabetics in Mexico. A cross-sectional study

PONE-D-23-03248R2

Dear Dr. Molina-Guarneros,

We’re pleased to inform you that your manuscript has been judged scientifically suitable for publication and will be formally accepted for publication once it meets all outstanding technical requirements.

Kind regards,

Tariq Jamal Siddiqi

Academic Editor

PLOS ONE
---

## [Editor Report · Acceptance letter]

18 Jul 2023

PONE-D-23-03248R2 

Prescription of oral antidiabetics in Mexico. A cross-sectional study 

Dear Dr. Molina-Guarneros:

I'm pleased to inform you that your manuscript has been deemed suitable for publication in PLOS ONE. Congratulations! Your manuscript is now with our production department. 

Kind regards, 

on behalf of

Dr. Tariq Jamal Siddiqi 

Academic Editor

PLOS ONE